# CASIAL: GEOMETRIC DISTORTION ROBUST IMAGE WATERMARKING

## ABSTRACT

Deep learning–based watermarking has shown strong robustness against non-geometric distortions, yet its performance under geometric transformations remains limited. Such transformations induce two fundamental failure modes: region removal, such as cropping or masking, which eliminates the information carried by removed pixels, and desynchronization, such as scaling or rotation, which misaligns pixel positions and disrupts decoding. We argue that achieving geometric robustness requires two essential properties: (1) global spread of the watermark message, ensuring resilience even when large regions are removed, and (2) geometry-invariant representations, enabling decoding to remain synchronized despite spatial transformations. To realize these properties, we propose CASIAL, a geometric distortion–robust watermarking framework with cover image-aware message spreading (CAS) and invariance alignment learning (IAL). CAS tightly couples watermark bits with cover image features and distributes them adaptively across the entire image, enhancing per-pixel information capacity and robustness to region removal. IAL leverages spatial attention to capture cross-pixel dependencies and align perturbed features into a shared geometry-invariant representation space, mitigating failures due to desynchronization. Extensive experiments demonstrate that CASIAL achieves state-of-the-art robustness against challenging geometric distortions, while maintaining high visual fidelity and decoding accuracy.

## 1 INTRODUCTION

Digital watermarking (Van Schyndel et al., 1994) has long been used to protect intellectual property for digital media (Zhu et al., 2018; Liu et al., 2025). Typically, it operates in two stages: a secret message is first embedded in the media and later extracted to verify authorship. Traditionally, watermarking has been applied primarily to human-captured or manually created content. However, with the rise of text-to-image (T2I) models (Ramesh et al., 2021; Saharia et al., 2022; Chen et al., 2024), high-quality image generation has become widely accessible, and malicious people can fabricate realistic yet misleading images to spread false information or fake news (Walker et al., 2024). As a result, digital watermarking is also widely used for tracing AI-generated images.

For watermarking methods, invisibility and robustness are the two most important properties. In deep learning–based watermarking models (Zhu et al., 2018; Zhang et al., 2021; Jia et al., 2021; Ma et al., 2022; Fang et al., 2023), robustness is mainly achieved via differentiable noise layers inserted during training, which guide the model to learn robust, distortion-invariant representations. Although recent watermarking frameworks ((Zhang et al., 2021; Jia et al., 2021; Ma et al., 2022; Fang et al., 2023)) achieve strong robustness against pixel-level noise, they remain fundamentally limited in addressing geometric transformations.

Unlike non-geometric distortions, which preserve spatial alignment, geometric transformations introduce two primary failure modes: **region removal** (e.g., cropping or masking), which removes subsets of pixels and thereby eliminates the information carried by those pixels, and **desynchronization** (e.g., scaling or rotation), which misaligns pixel positions and disrupts decoding. These challenges expose fundamental weaknesses in existing frameworks.

In particular, current models lack explicit mechanisms to ensure that the watermark message spreads across the entire image, leaving extraction fragile once local regions are removed. At the same time,

conventional CNN backbones, constrained by fixed receptive fields, struggle to perceive displaced structures or to align features within a geometry-invariant space. Together, these shortcomings highlight a key insight: achieving geometric robustness requires two complementary properties, global spread of the watermark message and the establishment of geometry-invariant representations.

To realize these properties, we propose **CASIAL**, a geometric distortion–robust watermarking framework with **c**over image-**a**ware message **s**preading (CAS) and **i**nvariance **a**lignment **l**earning (IAL). CAS tightly couples watermark bits with cover image features and distributes them across the entire representation, ensuring that the watermark message is redundantly and adaptively embedded throughout the image. IAL leverages spatial attention (Vaswani et al., 2017) to capture cross-pixel dependencies and align perturbed features into a shared geometry-invariant representation space, thereby preserving synchronization under spatial transformations.

Specifically, rather than processing message bits as independent data, CAS treats each bit as a control signal that selects between cover image–conditioned candidate features, so the resulting message features are derived directly from the cover image and each bit is naturally distributed across the entire image feature. This broad, cover-aware spreading mitigates failures caused by region removal. Meanwhile, IAL integrate basic spatial attention into both the backbone of the encoder and the decoder to enable adaptive reweighting of spatial regions: the model can dynamically reallocate attention to downweight corrupted or missing areas and upweight reliable content. Together, these components instantiate the two essential operations for geometric robustness, enabling CASIAL to withstand both region removal and desynchronization while preserving invisibility and fidelity.

In summary, our contributions are as follows:

- We identify that existing deep watermarking frameworks fall short under geometric transformations due to their inability to (i) broadly spread the watermark message across the image and (ii) align features into geometry-invariant representations. This analysis highlights the two essential properties required for geometric robustness.

- We propose CASIAL, a geometric distortion–robust watermarking framework that instantiates these two properties through (i) cover image-aware message spreading (CAS), which couples watermark bits with cover features and disperses them across the entire image, and (ii) invariance alignment learning (IAL), which employs spatial attention to align perturbed features into a geometry-invariant representation space.

- Extensive experiments under various geometric distortions demonstrate that CASIAL achieves state-of-the-art robustness while maintaining high visual fidelity.

## 2 RELATED WORKS

**After-generation watermarking** After-generation watermarking embeds a watermark directly into a image, independent of how the image was generated. To balance invisibility and robustness, traditional algorithms often operate in the transform domain (Daren et al., 2001; Ingemar et al., 2008), but hand-crafted designs limit adaptability to diverse distortions. Recent work therefore adopts trainable deep learning–based models, typically the Encoder–NoiseLayer–Decoder (END) framework (Zhu et al., 2018; Zhang et al., 2021; Jia et al., 2021), in which a noise layer is inserted during training to simulate distortions. The noise layer is central to robustness: differentiable approximations allow the encoder and decoder to learn distortion-invariant representations, and many studies design noise layers targeted to specific perturbations. For example, forward ASL (Zhang et al., 2021) models non-differentiable operations as additive noise to enable gradient backpropagation in non-differentiable noise layers, and MBRS (Jia et al., 2021) mixes simulated and real JPEG within a mini-batch to strengthen robustness to JPEG compression. Beyond digital perturbations, differentiable screen-to-camera (Tancik et al., 2020) and print-to-camera (Fang et al., 2022) noise layers have also been explored to improve robustness in physical scenarios. While these approaches are effective for many non-geometric distortions, geometric transformations disrupt spatial alignment and expose limitations that noise layers alone often cannot overcome.

**In-generation watermarking** The rise of text-to-image (T2I) models has attracted part of the community from after-generation watermarking to in-generation watermarking that integrates the watermark into the generative process. Among generative backbones, latent diffusion models

(LDMs) (Rombach et al., 2022; Peebles & Xie, 2023) are the most widely used due to their computational efficiency and high generated image quality, and thus have become the primary target for in-generation watermarking research. Around LDMs, two main lines have emerged. Fine-tuning–based methods (Fernandez et al., 2023) fine-tune the VAE decoder jointly with a pretrained extractor so that generated images carry extractor-detectable watermarks, with robustness still secured by training-time noise layers. In contrast, inversion-based methods (Yang et al., 2024; Li et al., 2025) avoid modifying model weights: they embed the watermark in the initial noise latent, generate the image via the standard denoise process, and at the decoding stage perform diffusion inversion to map the watermarked image back to the initial noise space for extraction. Their robustness is attributed in part to the add–remove Gaussian noise cycle of diffusion, which provides tolerance to Gaussian noise and some other non-geometric distortions even without explicit noise layers. However, inversion approaches typically rely on deterministic, ODE-style samplers (Song et al., 2021) that admit tractable inversion, limiting applicability under stochastic samplers or schedules.

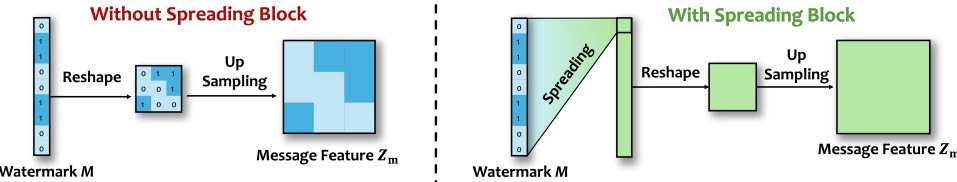

Figure 1: Two kinds of message processing blocks (MPBs). Left: without spreading block, the message is reshaped and directly upsampled. Right: with spreading block, a linear layer first spreads the message, followed by reshape and upsampling.

## 3 MOTIVATION AND METHODS

### 3.1 MOTIVATION

**Analysis of After-Generation Watermarking** For after-generation watermarking, robustness to geometric distortions cannot be secured by noise layers alone because the underlying vulnerability stems from structural deficiencies in both the encoder and the decoder. On the encoder side, the key weakness lies in the Message Processing Block (MPB). As shown in Fig. 1, existing MPBs typically fall into two groups. The first directly upsamples the message with transposed convolutions to match the spatial resolution of the image features before feature fusion (Jia et al., 2021; Fang et al., 2023). This confines each bit to localized regions, preventing effective distribution across the feature space and leaving the watermark vulnerable to region removal. The second group first projects the message with a linear layer, reshapes it, and then upsamples it with transposed convolutions (Jia et al., 2021; Ma et al., 2022). While this broadens the spread of information, the generated message features remain independent of the cover image features, leading to weak coupling and degraded visual quality of the watermarked image. On the decoder side, conventional CNN backbones use fixed kernel shapes and strides and cannot adapt to missing or misaligned regions caused by geometric distortions. As a result, the decoder is unable to reallocate attention away from corrupted areas toward informative areas, making reliable recovery difficult under region removal and spatial misalignment. These limitations suggest that a more effective encoder design should both spread message bits across the feature space and tightly couple them with cover image features, while the decoder should be endowed with the ability to adaptively focus on reliable spatial regions.

**Analysis of In-Generation Watermarking** For in-generation watermarking, existing approaches also face inherent limitations under geometric distortions. Fine-tuning-based methods (Fernandez et al., 2023) adapt the VAE decoder together with a pretrained watermark extractor, but the extractor itself is not robust to geometric transformations, and this weakness will be inherited by the generated watermark images. Inversion-based methods (Yang et al., 2024; Li et al., 2025) embed and extract watermarks in the initial noise space. To preserve image quality, the initial noise must retain the Gaussian distribution, which constrains the embedding process and limits the spread of messages. During extraction, the isotropic noise feature provides almost no structural cues for alignment, leaving the method vulnerable to geometric distortions. Despite GaussMarker (Li et al., 2025)

attempting to enforce synchronization in the noise space, progress remains limited and has been tested only under mild cropping and rotation distortions.

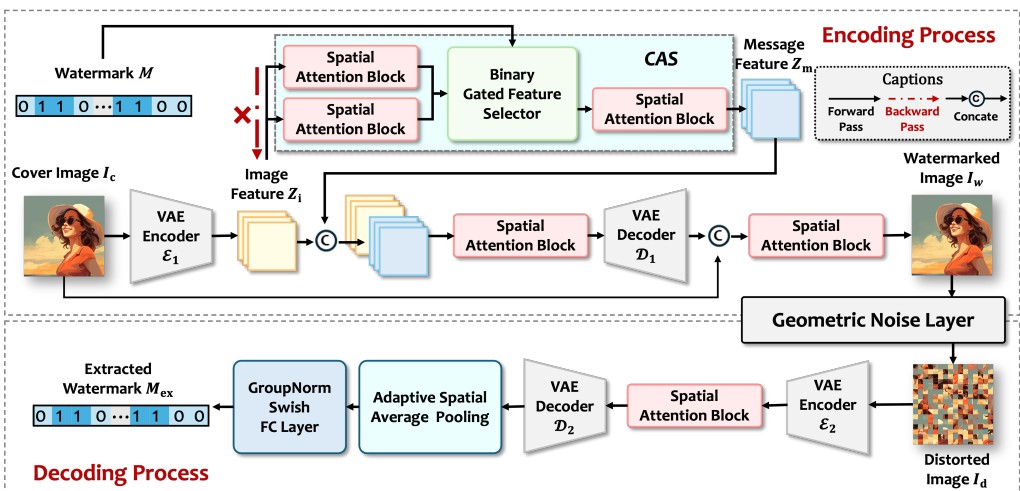

Figure 2: Overview of the proposed framework. In the encoder, a VAE encodes the cover image, CAS produces a cover-aware message feature, spatial attention fuses the image and message features, and a VAE decoder generates the watermarked image. During training, a noise layer introduces various distortions. In the decoder, a VAE encodes the distorted image, spatial attention refines the features, and a VAE decoder with a linear head recovers the message. Note that during backpropagation, gradients through CAS are used only to update this block and are not propagated into the image feature extractor.

## 3.2 METHODS

### 3.2.1 OVERVIEW

Our goal is to achieve robustness to geometric distortions by explicitly addressing two failure modes, region removal and desynchronization. As illustrated in Fig. 2, our model follows the Encoder-NoiseLayer-Decoder (END) framework and employs variational autoencoders (VAEs) (Rombach et al., 2022) for efficient feature extraction and fusion in latent space.

**Encoder** Given a cover image $I_c \in \mathbb{R}^{3 \times W \times H}$, the VAE encoder $\mathcal{E}_1$ transforms it to a latent cover image feature $Z_i = \mathcal{E}_1(I_c) \in \mathbb{R}^{C \times \frac{W}{4} \times \frac{H}{4}}$. Let $M \in \{-1, 1\}^L$ denote the binary secret message of length $L$. The CAS block takes $(Z_i, M)$ and outputs a message feature $Z_m$. At a high level it derives $Z_m$ directly from the cover image feature $Z_i$ under bit controlled selection. The encoder then fuses $Z_i$ and $Z_m$ using a spatial attention block $\tilde{Z} = \mathrm{SA}(Z_i \oplus Z_m)$, where $\oplus$ denotes channel concatenation. The VAE decoder $\mathcal{D}_1$ maps $\tilde{Z}$ back to image space, and a final spatial attention block refines the fusion with the cover image to yield the watermarked image $I_w = \mathrm{SA}\left(\mathcal{D}_1(\tilde{Z}) \oplus I_c\right)$.

**Noise layer** During training, the noise layer $\mathcal{N}$ applies simulated geometric distortions to the watermarked image to produce a distorted image $I_d = \mathcal{N}(I_w)$.

**Decoder** For the decoder, the input is a distorted image $I_d$. Another VAE encoder produces a latent feature $Z_i' = \mathcal{E}_2(I_d)$, which is then refined by a spatial attention block that adaptively extracts features by suppressing corrupted regions and emphasizing reliable content. The VAE decoder $\mathcal{D}_2$ then processes the refined features, followed by adaptive spatial average pooling and a linear layer that outputs the extracted watermark $M_{ex}$.

In summary, CAS tightly couples the secret bits with the cover-image feature via bit-guided selection, while spatial attention supplies adaptive spatial reweighting and global feature extraction to realign displaced structures. The noise layer simulates distortions during training and drives the model to learn robust, distortion-invariant representations.

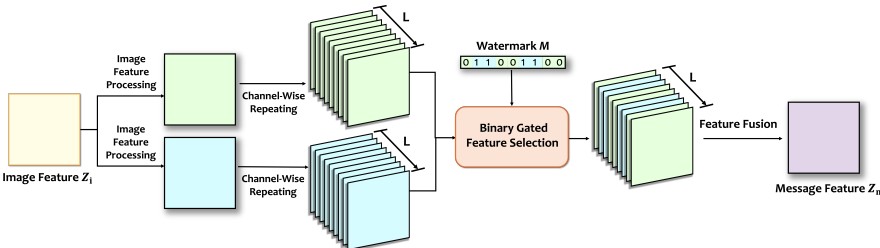

Figure 3: From the image feature $Z_\mathrm{i}$, two spatial-attention branches produce two candidate features for bit 0 and bit 1. Both candidate features are repeated $L$ times, and guided by the watermark bits, a binary-gated selector chooses one candidate per bit. The selected candidates are then fused to form the message feature $Z_\mathrm{m}$.

### 3.2.2 CAS Block

An overview of the CAS block is shown in Fig.3, and the algorithmic flow is presented in AppendixB.

**Binary candidate feature generation** The CAS block takes the latent cover image feature $Z_\mathrm{i}$ and the binary secret message $M$ as input. It first generates two candidate features from $Z_\mathrm{i}$ through two separate spatial attention blocks:

$$F^{(0)} = \mathrm{SA}_0(Z_\mathrm{i}), \qquad F^{(1)} = \mathrm{SA}_1(Z_\mathrm{i}),$$

where $F^{(0)}, F^{(1)} \in \mathbb{R}^{C \times \frac{H}{4} \times \frac{W}{4}}$ denote the candidate features for bit zero and bit one, respectively.

**Binary gated feature selection** The Binary Gated Feature Selector (BGFS) is a parameter free selector that treats each bit $m_k \in \{0, 1\}$ as a control signal. Given the two candidate features $F^{(0)}$ and $F^{(1)}$ and the binary message $M = [m_1, \ldots, m_L]$, we select for each bit $m_k$ a candidate feature as

$$S_k = (1 - m_k)\, F^{(0)} + m_k\, F^{(1)}.$$

Concatenate $\{S_k\}_{k=1}^{L}$ along the channel dimension to obtain

$$\tilde{S} = \mathrm{Concat}_{\mathrm{channel}}\big(S_1, \ldots, S_L\big) \in \mathbb{R}^{(L \cdot C) \times \frac{H}{4} \times \frac{W}{4}}.$$

**Attention-based message feature fusion** A spatial attention block fuses the concatenated channels in $\tilde{S}$ and produces the final message feature

$$Z_\mathrm{m} = \mathrm{SA}_2(\tilde{S}) \in \mathbb{R}^{C \times \frac{H}{4} \times \frac{W}{4}}.$$

Through selection, CAS uses an entire candidate feature to represent bit 0 or bit 1, so each bit is naturally spread over the whole image feature space. Moreover, bits act only as selection signals and every candidate feature is directly generated from the cover image features $Z_\mathrm{i}$. As a result, $Z_\mathrm{m}$ remains tightly coupled to the cover image features, jointly addressing the locality and weak-coupling issues of prior message processing blocks.

### 3.2.3 Loss Function

**Image Loss** During the encoding stage, the encoder takes the cover image $I_\mathrm{c}$ and the watermark $M$ as inputs and outputs the watermarked image $I_\mathrm{w}$. To keep the visual quality of $I_\mathrm{w}$ close to $I_\mathrm{c}$, the image loss is defined as follows:

$$\mathcal{L}_{\mathrm{Image}} = \mathcal{L}_{\mathrm{MSE}}(I_\mathrm{c}, I_\mathrm{w}). \tag{1}$$

**Message Loss** During the decoding stage, the decoder takes the distorted image $I_\mathrm{d} = \mathcal{N}(I_\mathrm{w})$ as input and predicts the extracted watermark $M_{\mathrm{ex}}$. To ensure robust and accurate decoding, the message loss is defined as follows:

$$\mathcal{L}_{\mathrm{Message}} = \mathcal{L}_{\mathrm{MSE}}(M, M_{\mathrm{ex}}). \tag{2}$$

**Total Loss** Visual quality and decoding accuracy present a trade off. The total loss is a weighted sum of the two losses:

$$\mathcal{L}_{\mathrm{Total}} = \lambda_1\, \mathcal{L}_{\mathrm{Image}} + \lambda_2\, \mathcal{L}_{\mathrm{Message}}, \tag{3}$$

where $\lambda_1$ and $\lambda_2$ balance the trade off between visual quality and decoding accuracy.

| Original Image | Crop&Resize | Erasing | Jigsaw | Elastic | Shear | Rotate |

Figure 4: Six geometric distortions used in our experiments: Crop & Resize, Erasing, Jigsaw distortion, Elastic deformation, Shear, and Rotation.

## 4 EXPERIMENTAL RESULTS

### 4.1 EXPERIMENTAL SETTINGS

**Dataset and Settings** We use the COCO dataset (Lin et al., 2014) for training. For after-generation watermarking, evaluation on real images is conducted on the USC SIPI image dataset (Viterbi, 1977). For in-generation methods, testing images are generated with prompts from Stable Diffusion Prompts (Gustavosta, 2022) following the settings of Gaussian Shading (Yang et al., 2024) and GaussMarker (Li et al., 2025). In after-generation experiments the cover image has $C = 3$, $W = 128$, $H = 128$, and the message length is $L = 64$. For in-generation experiments we follow Gaussian Shading and GaussMarker with $C = 3$, $W = 512$, $H = 512$, and the message length $L = 64$. To comprehensively evaluate robustness to geometric distortions, we test six representative cases: crop & resize (cropping a region and rescaling it to the original resolution), erasing, jigsaw distortion (partitioning the image into sub-blocks and randomly permuting them), elastic deformation, shear, and rotation. The effects of these distortions are illustrated in Fig. 4. The image loss weight $\lambda_1$ and the message loss weight $\lambda_2$ are both set to 1 at the start of training. We use the Adam optimizer (Kingma & Ba, 2015) with learning rate $1 \times 10^{-5}$ and default hyperparameters. All models are trained on two NVIDIA RTX A40 GPUs.

**Benchmarks** To comprehensively assess robustness to geometric distortions, we evaluate five after-generation watermarking methods: dwtDctSvd (Ingemar et al., 2008), CIN (Ma et al., 2022), FIN (Fang et al., 2023), MBRS, and MBRS with a spreading module (Jia et al., 2021). We also evaluate two in-generation methods: Gaussian Shading (Yang et al., 2024) and GaussMarker (Li et al., 2025) with Stable Diffusion V2.1.

**Evaluation metrics** For robustness, we report decoding accuracy (ACC), where higher values indicate better performance. For after-generation watermarking, we measure visual quality using peak signal to noise ratio (PSNR). For in-generation watermarking, we use FID (Heusel et al., 2017) and CLIP Score (Radford et al., 2021) as visual metrics.

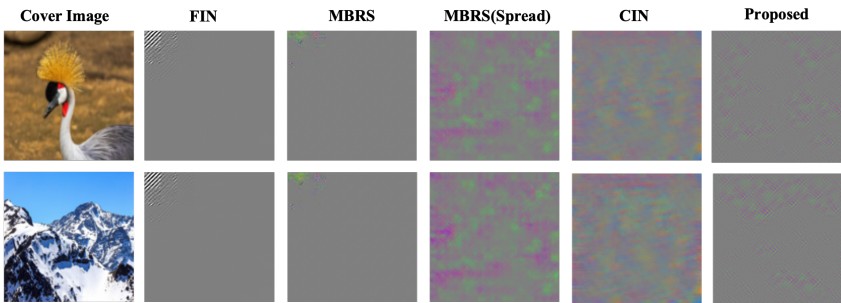

| Cover Image | FIN | MBRS | MBRS(Spread) | CIN | Proposed |

Figure 5: For each method and cover image, we embed (i) an all-zero message and (ii) the same message with only the first bit flipped to 1, then visualize the absolute difference between the two watermarked images. Image-wide residuals indicate good bit spreading, while localized blobs indicate poor spread. Residual patterns that vary with the cover image imply strong content coupling.

### 4.2 ANALYSIS OF ENCODERS

In Section 3.1 we analyze the limitations in two categories of message processing blocks (MPBs). We now validate this analysis experimentally. As shown in Fig. 5, FIN and MBRS, which upsample the message $M$ directly, fail to spread the first bit across the full image. By contrast, MBRS (Spread)

and CIN use a linear projection before upsampling achieve broader spreading. However, all four models exhibit weak coupling to the cover image: the resulting difference maps are nearly identical across two different cover images, indicating a lack of content adaptivity that may degrade visual quality. In comparison, our approach spreads each bit over the entire image feature while get the watermark feature directly from the cover image, thus maintaining strong coupling to image content. This confirms the advantages of the CAS block in global bit spreading and cover image coupling ability.

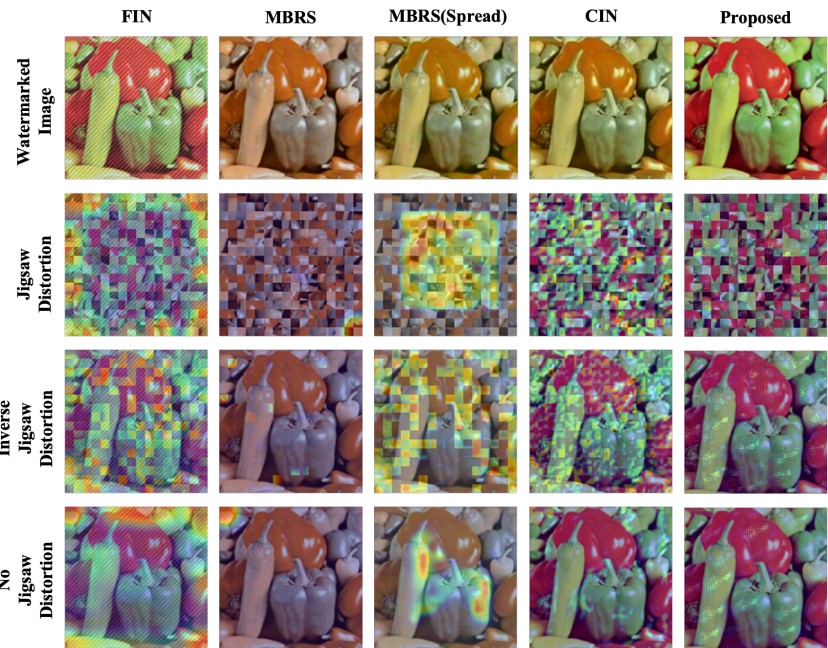

Figure 6: Decoder attention under Jigsaw distortion. Rows (top to bottom): watermarked image; attention on the jigsaw-distorted watermarked image; inverse-permuted attention obtained by applying the inverse permutation to row 2; attention on the original watermarked image. Large discrepancies between rows 3 and 4 indicate poor realignment; our decoder's row 3 closely matches row 4, evidencing robust resynchronization.

## 4.3 ANALYSIS OF DECODERS

When geometric distortions cause information loss or spatial displacement, a robust decoder must attend globally and reallocate attention adaptively to learn distortion-invariant representations. To make this concrete, we visualize the spatial attention map from the decoder under a jigsaw distortion in Fig. 6. For each method, we present four rows: the original watermarked image, the attention map on the jigsaw distorted watermarked image, the same attention map inverse permuted back to the original coordinates, and the attention map on the undistorted watermarked image. We refer to the attention on the undistorted input as the clean attention. Correct decoding requires that, on the distorted input, the decoder can still redistribute attention to the corresponding regions that are highlighted in the clean attention. Baseline decoders show large discrepancies between the inverse permuted attention and the clean attention, indicating failure to relocate the correct regions after spatial displacemen. In contrast, our decoder produces an inverse permuted attention that closely matches the clean attention, demonstrating effective realignment and robustness under severe desynchronization.

## 4.4 COMPARISON TO BASELINES

For the after-generation comparison, we retrain FIN, MBRS, MBRS (Spread), and CIN using the official implementations under identical noise-layer settings. We first train one model per distortion to isolate effects. We then train a single model on the mixture of all six distortions and evaluate it across all distortions. For the in-generation comparison, we use our mixed-distortion model. Comparative experiments on non-geometric distortions are provided in Appendix C.

### 4.4.1 COMPARISON TO AFTER-GENERATION WATERMARKING

Table 1: Single distortion training: invisibility (PSNR, dB) and robustness (decoding accuracy, %) on six geometric distortions. MBRS (Spread) denotes MBRS with a spreading block.

| Model | Shear (%) | | | Rotation (%) | | | Elastic (%) | | |
|---|---|---|---|---|---|---|---|---|---|
| | PSNR(dB)↑ | d = 75 | 80 | PSNR(dB)↑ | d = 150 | 180 | PSNR(dB)↑ | α = 4 | 6 |
| dwtDctSvd | 35.49 | 49.90 | 49.79 | 35.49 | 49.81 | 50.11 | 35.49 | 53.80 | 51.78 |
| FIN | 33.55 | 69.43 | 64.84 | 31.67 | 51.86 | 48.83 | 24.12 | 96.58 | 84.77 |
| MBRS | 38.75 | 64.84 | 56.05 | 45.71 | 64.26 | 51.46 | 52.19 | 97.85 | 93.85 |
| MBRS(Spread) | 44.17 | 92.19 | 72.46 | 57.36 | 99.35 | 99.43 | 59.10 | 99.80 | 98.59 |
| CIN | 49.25 | 99.61 | 98.12 | 37.93 | 99.31 | 99.51 | 42.19 | 93.54 | 85.84 |
| Proposed | **51.12** | **99.92** | **98.98** | **59.02** | **99.51** | **99.78** | **60.65** | **99.99** | **99.21** |

| Model | Jigsaw (%) | | | C&R (%) | | | Erase (%) | | |
|---|---|---|---|---|---|---|---|---|---|
| | PSNR(dB)↑ | g = 16 | 32 | PSNR(dB)↑ | r = 0.05 | 0.01 | PSNR(dB)↑ | r = 0.95 | 0.99 |
| dwtDctSvd | 35.49 | 51.52 | 50.52 | 35.49 | 49.90 | 49.90 | 35.49 | 49.97 | 49.85 |
| FIN | 29.68 | 56.42 | 52.93 | 29.26 | 49.41 | 49.52 | 28.34 | 81.25 | 72.85 |
| MBRS | 37.09 | 62.72 | 62.11 | 39.07 | 54.19 | 51.67 | 35.61 | 82.71 | 71.09 |
| MBRS(Spread) | 43.73 | 61.62 | 61.23 | 39.60 | 55.59 | 50.49 | 56.45 | 99.87 | 99.36 |
| CIN | 44.41 | 82.86 | 63.96 | 39.18 | 51.14 | 51.07 | 55.38 | 99.61 | 98.32 |
| Proposed | **48.87** | **99.97** | **99.61** | **41.03** | **100** | **99.01** | **57.21** | **100** | **99.69** |

**Single Distortion Training** The noise layer is configured as follows: shear with angle sampled from $[-80°, 80°]$; rotation from $[-180°, 180°]$; elastic transformation with scaling factor equals to 4; jigsaw with grid size drawn from $\{8, 16, 32\}$; crop and resize with crop ratio sampled from $[0.01, 1]$ followed by resizing to the original resolution; and erasing with area ratio within $[0, 0.99]$. Table 1 shows that our method attains the best visual quality (PSNR) and decoding accuracy across all six distortions. Competing models achieve comparable results on shear, rotation, elastic, and erasing at moderate severity, but they totally fail under the two critical failure modes: C&R (region removal) and jigsaw (desynchronization). In contrast, our approach sustains near-perfect accuracy in these extreme cases, consistent with the benefits of tightly coupling message features with cover image features and employing spatial attention for adaptive realignment.

Table 2: Joint training on six geometric distortions: invisibility (PSNR, dB) and robustness (decoding accuracy, %) with average accuracy across distortions. MBRS (Spread) denotes MBRS with a spreading block.

| Method | PSNR↑ (dB) | Shear (%) | Rotation (%) | Elastic (%) | Jigsaw (%) | C&R (%) | Erase (%) | **Ave** (%) |
|---|---|---|---|---|---|---|---|---|
| dwtDctSvd | 35.49 | 50.02 | 50.11 | 53.80 | 51.52 | 49.93 | 51.23 | 51.10 |
| FIN | 25.89 | 50.49 | 50.20 | 99.04 | 49.71 | 51.46 | 95.31 | 66.04 |
| MBRS(Spread) | 27.27 | 58.13 | 53.45 | 75.49 | 50.09 | 53.22 | 63.18 | 58.93 |
| MBRS | 26.09 | 61.15 | 50.80 | 98.94 | 51.52 | 52.36 | 77.86 | 65.41 |
| CIN | 25.05 | 97.56 | 88.48 | 99.13 | 63.21 | 54.21 | 99.01 | 83.47 |
| Proposed | **52.03** | **99.99** | **99.98** | **100** | **100** | **99.66** | **99.94** | **99.93** |

**Joint Training on Six Distortions** We train a single model with a mixed pool of distortions using the following settings: shear with angle sampled from $[-60°, 60°]$; rotation from $[-180°, 180°]$; elastic transformation with scaling factor equals to 2; jigsaw with grid size equals to 16; crop and resize with crop ratio from $[0.1, 1]$ followed by resizing to the original resolution; and erasing with area ratio from $[0, 0.9]$. As shown in Table 2, under joint training the after-generation baselines not only continue to fail on C&R and jigsaw, but also suffer substantial robustness drops on the other geometric distortions, together with noticeable PSNR degradation. This indicates that differentiable geometric noise layers alone are insufficient to yield geometric robustness given existing encoder and decoder designs. In contrast, our method maintains near-perfect decoding across all six distortions while preserving the highest PSNR.

### 4.4.2 COMPARISON TO IN-GENERATION WATERMARKING

We evaluate against two in-generation baselines using the model from our *Joint Training on Six Distortions* setting. As shown in Table 3, our method closely matches the clean "No Watermark" image quality while achieving 100% decoding accuracy under all six geometric distortions. Gauss-Marker improves over Gaussian Shading on rotation, elastic, and erasing, but still fails on shear,

Table 3: Benchmark comparisons: image quality (CLIP Score, FID) and robustness (decoding accuracy, %) under six geometric distortions, with average accuracy across distortions. Gauss Shading abbreviates Gaussian Shading.

| Method | CLIP Score↑ | FID↓ | Shear (%) | Rotation (%) | Elastic (%) | Jigsaw (%) | C&R (%) | Erase (%) | Ave (%) |
|---|---|---|---|---|---|---|---|---|---|
| No Watermark | **0.3645** | **25.31** | – | – | – | – | – | – | – |
| Gauss Shading | 0.3637 | 25.49 | 52.22 | 49.64 | 79.64 | 51.03 | 50.64 | 78.14 | 60.22 |
| GaussMarker | 0.3642 | 25.99 | 66.45 | 97.63 | 95.15 | 56.11 | 53.25 | 98.45 | 77.84 |
| Proposed | 0.3642 | 25.39 | **100** | **100** | **100** | **100** | **100** | **100** | **100** |

jigsaw, and C&R. These results demonstrate the competitiveness of our approach for watermarking AI-generated images, providing substantially stronger robustness to geometric distortions than existing in-generation watermarking methods. Compared with the after-generation results from Table 2, part of the gain is explained by a lower bits-per-pixel (bpp) payload: the in-generation experiments use $512 \times 512$ images with the same message length $L = 64$, increasing the spatial support per bit by $16\times$ relative to $128 \times 128$. This additional redundancy benefits CAS, helping preserve visual quality and achieve better robustness across all distortions.

## 4.5 ABLATION STUDY

Table 4: Ablation: invisibility (PSNR, dB) and robustness (decoding accuracy, %) under six geometric distortions, with average accuracy across distortions. SP denotes the Spatial Attention block.

| Method | PSNR↑ (dB) | Shear (%) | Rotation (%) | Elastic (%) | Jigsaw (%) | C&R (%) | Erase (%) | Ave (%) |
|---|---|---|---|---|---|---|---|---|
| w/o SP | 46.98 | 82.18 | 99.96 | 99.88 | 99.97 | 92.82 | 87.72 | 94.64 |
| w/o CAS | 37.92 | 78.40 | 86.69 | 96.87 | 92.75 | 72.19 | 87.43 | 84.77 |
| Proposed | **52.03** | **99.99** | **99.98** | **100** | **100** | **99.66** | **99.94** | **99.93** |

**Importance of Spatial Attention and CAS.** To assess the roles of spatial attention and the proposed CAS, we conduct two ablations in Table 4. In w/o SP (Spatial Attention), we remove all spatial attention blocks from both the encoder (including those inside CAS) and the decoder. In w/o CAS, we keep the rest of the architecture unchanged but replace CAS with the message processing block from MBRS (Spread). From the table, we can find two observations. First, w/o CAS suffers the broadest degradation in both PSNR and accuracy across all six distortions, indicating that bit-guided selection from cover-conditioned candidate features is the principal driver of robustness and invisibility, consistent with CAS's role in spreading each bit across the entire feature region while keeping it coupled with image content. Second, w/o SP retains some gains from CAS but still shows clear drops, most visibly under heavy region removal and strong shear. This pattern suggests that spatial attention is key for adaptively reweighting reliable regions and helping the decoder reallocate attention when structures are displaced or partially missing.

## 5 CONCLUSION

This paper strengthens the geometric robustness of deep learning–based watermarking. We propose CAS (cover image-aware message spreading), which derives message features from cover image features via bit-guided selection, yielding cover image coupled, spatially distributed representations that resist region removal and desynchronization. Within the END framework, we also integrate spatial attention into both the encoder and the decoder to adaptively reweight spatial regions under misalignment. Extensive experiments across six geometric distortions show consistent gains in decoding accuracy and visual quality over after-generation and in-generation baselines. Ablation studies confirm the effectiveness of our design: CAS provides cover-image coupling and broad bit spreading, while spatial attention supplies global aggregation and adaptive attention reallocation, mitigating region removal and desynchronization.

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

## A  LLM Usage

Large Language Models (LLMs) were used to aid in the writing and polishing of the manuscript. Specifically, we used an LLM to assist in refining the language, improving readability, and ensuring clarity in various sections of the paper. The model helped with tasks such as sentence rephrasing, grammar checking, and enhancing the overall flow of the text.

It is important to note that the LLM was not involved in the ideation, research methodology, or experimental design. All research concepts, ideas, and analyses were developed and conducted by the authors. The contributions of the LLM were solely focused on improving the linguistic quality of the paper, with no involvement in the scientific content or data analysis.

The authors take full responsibility for the content of the manuscript, including any text generated or polished by the LLM. We have ensured that the LLM-generated text adheres to ethical guidelines and does not contribute to plagiarism or scientific misconduct.

## B  CAS Algorithm

---

**Algorithm 1** CAS: Cover Image Aware Message Spreading

---

**Input:** Cover-image feature $Z_{\text{i}} \in \mathbb{R}^{C \times \frac{H}{4} \times \frac{W}{4}}$ ; binary message $M = [m_1, \ldots, m_L] \in \{0,1\}^L$

**Output:** Message feature $Z_{\text{m}} \in \mathbb{R}^{C \times \frac{H}{4} \times \frac{W}{4}}$

1: **Candidate feature generation**
2: $F^{(0)} \leftarrow \text{SA}_0(Z_{\text{i}})$          ▷ candidate for bit 0
3: $F^{(1)} \leftarrow \text{SA}_1(Z_{\text{i}})$          ▷ candidate for bit 1

4: **Binary gated feature selection (BGFS)**
5: **for** $k = 1$ to $L$ **do**
6:      $S_k \leftarrow (1 - m_k) F^{(0)} + m_k F^{(1)}$          ▷ $S_k \in \mathbb{R}^{C \times \frac{H}{4} \times \frac{W}{4}}$
7: **end for**
8: $\tilde{S} \leftarrow \text{Concat}_{\text{channel}}(S_1, \ldots, S_L)$          ▷ $\tilde{S} \in \mathbb{R}^{(L \cdot C) \times \frac{H}{4} \times \frac{W}{4}}$

9: **Attention-based feature fusion**
10: $Z_{\text{m}} \leftarrow \text{SA}_2(\tilde{S})$
11: **return** $Z_{\text{m}}$

---

## C  Baseline Comparison under Non-Geometric Distortions

### C.1  Comparison to After-Generation Watermarking

**Single Distortion Training** The noise layer is configured as follows: Gaussian Blur (GB) with a standard deviation of 2.0 and a kernel size of 7, Median Blur (MB) with a kernel size of 7, Gaussian Noise (GN) with a variance of 0.05 and a mean of 0, Salt & Pepper Noise (S&P) with a noise ratio of 0.1, Dropout (DP) with a drop ratio of 0.6, JPEG Compression (JPEG) with a quality factor of 50, and JPEGSS (simulated differentiable JPEG) with a quality factor of 50. Table 5 shows that our method attains the best or second-best PSNR and decoding accuracy across all non-geometric distortions: it leads on S&P, JPEG, dropout, and Gaussian blur, closely matches the top accuracy under Gaussian noise, and is slightly behind MBRS(Spread) on median blur. Overall, CAS preserves high visual quality while delivering state-of-the-art robustness to non-geometric perturbations.

**Joint Training on Six Distortions** We train a single model with a mixed pool of distortions using the settings same as above. Testing on Gaussian Blur (GB) with a standard deviation of 2.0 and a kernel size of 7, Median Blur (MB) with a kernel size of 7, Gaussian Noise (GN) with a variance of 0.01 and a mean of 0, Salt & Pepper Noise (S&P) with a noise ratio of 0.1, Dropout (DP) with a drop ratio of 0.6, JPEG Compression (JPEG) with a quality factor of 50. As shown in Table 6, our method achieves the highest PSNR (41.77,dB) and the best average decoding accuracy

Table 5: Single distortion training: invisibility (PSNR, dB) and robustness (decoding accuracy, %) on six non-geometric distortions. MBRS (Spread) denotes MBRS with a spreading block.

| Model | S&P Noise (%) | | | Gaussian Noise (%) | | | JPEG Compression (%) | | |
|---|---|---|---|---|---|---|---|---|---|
| | PSNR(dB)↑ | r = 0.09 | 0.10 | PSNR(dB)↑ | var = 0.04 | 0.05 | PSNR(dB)↑ | QF = 50 | 60 |
| dwtDctSvd | 35.49 | 73.39 | 69.74 | 35.49 | 52.42 | 50.31 | 35.49 | 68.26 | 75.81 |
| FIN | 63.53 | 99.25 | 99.02 | 40.35 | 99.41 | 99.04 | 48.76 | 98.24 | 99.51 |
| MBRS | 67.73 | 99.91 | 99.89 | **40.36** | 99.53 | 99.05 | 47.82 | 96.01 | 97.85 |
| MBRS(Spread) | 68.79 | 99.99 | 99.99 | 40.21 | **99.86** | **99.12** | 48.95 | 99.12 | 99.92 |
| CIN | 66.31 | 97.41 | 97.26 | 39.77 | 99.33 | 98.42 | 48.47 | 84.77 | 88.58 |
| Proposed | **69.14** | **100** | 99.99 | 40.01 | 99.76 | 99.08 | **49.31** | 99.31 | **99.97** |

| Model | Dropout (%) | | | Gaussian Blur (%) | | | Median Blur (%) | | |
|---|---|---|---|---|---|---|---|---|---|
| | PSNR(dB)↑ | r = 0.60 | 0.50 | PSNR(dB)↑ | σ = 1 | 2 | PSNR(dB)↑ | w = 5 | 7 |
| dwtDctSvd | 35.49 | 74.68 | 85.66 | 35.49 | 96.76 | 87.28 | 35.49 | 90.49 | 83.00 |
| FIN | 62.58 | 99.22 | 99.61 | 52.51 | 99.80 | 97.27 | 41.97 | 99.54 | 99.13 |
| MBRS | 70.83 | 99.05 | 99.51 | 65.78 | 99.84 | 99.21 | 51.31 | 98.25 | 98.32 |
| MBRS(Spread) | 69.59 | 95.13 | 97.23 | 65.82 | 99.98 | 98.22 | **55.30** | 99.90 | 99.50 |
| CIN | 63.41 | 97.07 | 98.73 | 63.32 | 98.83 | 97.66 | 49.08 | 98.86 | 98.44 |
| Proposed | **72.72** | **99.70** | **99.97** | **68.89** | **100** | **99.93** | 54.77 | 99.81 | 99.24 |

Table 6: Joint training on six non-geometric distortions: invisibility (PSNR, dB) and robustness (decoding accuracy, %) with average accuracy across distortions. MBRS (Spread) denotes MBRS with a spreading block.

| Method | PSNR↑ (dB) | JPEG (%) | MB (%) | GB (%) | S&P (%) | GN (%) | Dropout (%) | Ave (%) |
|---|---|---|---|---|---|---|---|---|
| dwtDctSvd | 35.49 | 68.26 | 83.00 | 87.28 | 69.74 | 78.10 | 74.68 | 76.84 |
| FIN | 41.58 | 99.90 | 99.56 | 99.97 | 99.99 | 99.24 | 99.80 | 99.74 |
| MBRS | 40.72 | 98.85 | 99.38 | 99.89 | 99.95 | 99.69 | 99.76 | 99.54 |
| MBRS(Spread) | 40.45 | 95.45 | **99.92** | 99.69 | 99.99 | **100** | 99.99 | 99.16 |
| CIN | 40.31 | 86.01 | 99.58 | 99.84 | **100** | 99.73 | 99.78 | 97.49 |
| Proposed | **41.77** | **99.96** | 99.86 | **100** | 99.99 | 99.98 | 99.99 | **99.96** |

(99.96%), with near-ceiling performance across all six distortions. FIN and the MBRS family (including MBRS (Spread)) reach high accuracy on several cases, but their averages and PSNRs are lower; the classical dwtDctSvd lags by a wide margin. These results indicate that our framework is also robust non-geometric distortions.

### C.1.1 COMPARISON TO IN-GENERATION WATERMARKING

Table 7: Benchmark comparisons: image quality (CLIP Score, FID) and robustness (decoding accuracy, %) under six geometric distortions, with average accuracy across distortions. Gauss Shading abbreviates Gaussian Shading.

| Method | CLIP Score↑ | FID↓ | JPEG (%) | MB (%) | GB (%) | S&P (%) | GN (%) | Ave (%) |
|---|---|---|---|---|---|---|---|---|
| No Watermark | **0.3645** | **25.31** | – | – | – | – | – | |
| Gauss Shading | 0.3637 | 25.49 | 99.90 | **99.97** | 99.97 | 96.80 | 96.38 | 98.60 |
| GaussMarker | 0.3642 | 25.99 | 98.61 | 98.54 | 98.55 | 98.25 | 99.47 | 98.68 |
| Proposed | 0.3640 | 25.68 | **99.93** | 99.94 | 99.85 | **99.92** | **99.94** | 99.92 |

We evaluate against two in-generation baselines using the model from our *Joint Training on Six Distortions* setting. As shown in Table 7, our method matches clean image quality closely (CLIP 0.3640 vs. 0.3645, FID 25.68 vs. 25.31) while achieving the best average robustness. GaussMarker and GaussShading are competitive on several distortions but trail on S&P and GN, resulting in lower averages (98.68% and 98.60%, respectively). These results indicate that our approach preserves visual quality and delivers good non-geometric robustness in the in-generation setting.

