# OpenReview forum: "CASIAL: Geometric Distortion Robust Image Watermarking"
_ICLR.cc/2026/Conference — Submitted to ICLR 2026_

### Official Review · Reviewer_x28o · 2025-10-25

**Soundness:** 2
**Presentation:** 1
**Contribution:** 2
**Rating:** 4
**Confidence:** 2

**Summary:**

This paper presents CASIAL, a deep-learning-based watermarking framework designed to enhance robustness against geometric distortions such as cropping, rotation, shear, and elastic deformation. The method introduces two core components: Cover Image-Aware Spreading (CAS), which tightly couples watermark bits with image features and distributes them globally, and Invariance Alignment Learning (IAL), which employs spatial attention to establish geometry-invariant representations. CASIAL is evaluated on six geometric distortion types and compared against prior after-generation and in-generation watermarking models such as MBRS, CIN, and GaussMarker. The model consistently achieves higher decoding accuracy (≈100%) and visual fidelity (PSNR > 50dB) across all tested distortions, demonstrating strong resistance to both region removal and desynchronization effects.

**Strengths:**

- Clear Identification of Geometric Weaknesses : The paper clearly articulates why existing watermarking methods fail under geometric transformations—namely, due to localized bit embedding and lack of spatial alignment mechanisms.
- Novel Dual-Mechanism Design (CAS + IAL) : The introduction of CAS for bit spreading and IAL for spatial attention alignment forms a well-motivated and complementary framework. This addresses both coverage and desynchronization, offering a practical path toward geometric robustness.
- Strong Empirical Performance : CASIAL demonstrates state-of-the-art robustness across all six geometric distortions (crop, erase, shear, rotation, elastic, and jigsaw) with near-perfect decoding accuracy, outperforming classical (DWT-DCT-SVD) and deep (MBRS, CIN) baselines.
- Comprehensive Experimental Setup : The paper performs both single-distortion and mixed-distortion training, as well as comparisons with in-generation watermarking methods. The results are numerically consistent and well-controlled.
- Thorough Ablation Studies : The contribution of each module—CAS and spatial attention—is well-isolated through ablations, supporting the authors’ design claims.

**Weaknesses:**

- Questionable Practical Motivation for Geometric Distortions : The relevance of extreme geometric transformations (e.g., heavy shear, jigsaw permutation) is unclear in real-world watermark removal scenarios. It is debatable whether users or attackers would intentionally apply such transformations before reusing an image.
- Limited Qualitative Evidence : Despite detailed quantitative results, the paper lacks visual comparisons of watermarked vs. distorted images. Without qualitative samples, it is difficult to judge perceptual quality and whether the high PSNR translates to visual fidelity.
- Dataset Scope and Relevance : The use of COCO and USC SIPI datasets provides general coverage but may not reflect realistic watermarking domains such as generative model outputs, photographic images, or artistic content.
- Inaccessible Supplementary Results : The paper references Tables 1 and 2 for key comparisons but does not provide the actual watermarked and decoded image examples. The absence of qualitative results in the supplementary materials limits reproducibility and visual verification.
- Overemphasis on Synthetic Benchmarks : While the performance on artificial distortions is impressive, real-use scenarios like printing–scanning or social media recompression are not explored. This weakens claims of real-world robustness.

**Questions:**

- Motivation for Geometric Robustness : In what realistic scenarios would geometric distortions (shear, jigsaw, or elastic deformation) be deliberately applied to remove a watermark? Could you clarify practical motivations beyond synthetic benchmarks?
- Qualitative Visualization : Could the authors provide visual comparisons between watermarked and distorted images to demonstrate perceptual quality? Are these missing due to confidentiality or dataset licensing issues?
- Access to Supplementary Files : The tables summarize extensive results, but corresponding image samples are absent. Will these be released upon acceptance, and do they reflect all distortion types?
- Extension to Real-World Scenarios : Has CASIAL been tested under camera-captured, printed, or recompressed (JPEG/web) distortions? How does it perform in such uncontrolled physical or platform-based environments?
- Model Generalization : Given the use of synthetic COCO training data, how does CASIAL generalize to unseen natural images or generative images from diffusion models?

---

### Official Review · Reviewer_LVis · 2025-10-30

**Soundness:** 2
**Presentation:** 3
**Contribution:** 2
**Rating:** 4
**Confidence:** 4

**Summary:**

The paper proposes CASIAL, a deep learning-based image watermarking framework that is robust against geometric distortions. The authors point out that existing methods perform well under pixel-level noise, but tend to fail under geometric distortions due to uneven information distribution and difficulties in feature alignment during decoding. To address this, CASIAL introduces two main innovations:
- CAS: This mechanism adaptively couples the watermark information with image features and distributes it broadly and redundantly throughout the entire image, thereby improving robustness against region removal (such as cropping).
- IAL: By incorporating spatial attention mechanisms into both the encoder and decoder, this module aligns perturbed features into a geometry-invariant space, enhancing resistance to desynchronization caused by distortions such as rotation and scaling.

Experimental results show that CASIAL achieves state-of-the-art robustness under various geometric distortions, while maintaining high visual quality and decoding accuracy.

**Strengths:**

1. The paper is clearly written and articulates the core issue of the difficulty in ensuring watermark robustness under geometric distortions.
2. the introduction of spatial attention modules enables adaptive spatial reweighting and global feature aggregation during both encoding and decoding stages, enhancing the model’s ability to handle geometric perturbations such as cropping and rotation. The binary-gated feature selection in the CAS block is relatively novel and demonstrates a certain degree of innovation.

**Weaknesses:**

1. Although the proposed CASIAL framework demonstrates some innovation in message spreading and coupling with cover features, there are still several significant shortcomings in the current presentation and methodology. Firstly, the novelty of the spatial attention mechanism is rather limited, as similar attention modules have already been widely adopted in recent deep learning tasks[1,2]. The paper does not clearly explain how its use of spatial attention is fundamentally different from or superior to existing methods.

2. Although the paper describes the functions and structures of each module in the network, these descriptions are overly brief, and more detailed explanations of each module’s structure and role would be beneficial.

[1] Woo S, Park J, Lee J Y, et al. Cbam: Convolutional block attention module[C]//Proceedings of the European conference on computer vision (ECCV). 2018: 3-19.

[2] Yuan S, Qin H, Yan X, et al. Sctransnet: Spatial-channel cross transformer network for infrared small target detection[J]. IEEE Transactions on Geoscience and Remote Sensing, 2024, 62: 1-15.

**Questions:**

1. How does the method perform under compounded or real-world distortions, such as combinations of geometric and non-geometric attacks, or distortions introduced by printing and photographing?
2. In what ways, if any, does the spatial attention mechanism in this framework differ from or improve upon standard usage in recent watermarking models?

---

### Official Review · Reviewer_Dohw · 2025-11-02

**Soundness:** 3
**Presentation:** 3
**Contribution:** 3
**Rating:** 6
**Confidence:** 3

**Summary:**

This paper proposes CASIAL, a geometric-distortion-robust image watermarking framework. The authors identify structural weaknesses of existing deep-learning-based watermarking under region removal and desynchronization, and introduce two core modules to address them: Cover image-Aware message Spreading (CAS), which uses bit-guided candidate-feature selection to tightly couple watermark bits with cover image features and adaptively distribute them across the whole image, and Invariance Alignment Learning (IAL), which injects spatial attention into the encoder/decoder to align perturbed features into a geometry-invariant representation space and restore synchronization. The paper evaluates CASIAL against multiple after-/in-generation baselines on six representative geometric distortions, reporting substantial improvements in decoding accuracy (ACC) and visual quality metrics (PSNR / FID / CLIP). Ablation studies confirm the importance of CAS and spatial attention.

**Strengths:**

1.The paper explicitly identifies two fundamental failure modes, region removal and desynchronization, and motivates why existing END-style approaches and conventional CNN backbones struggle under these modes.

2.CAS treats each bit as a control signal to select cover-conditioned candidate features, achieving global, cover-aware spreading; IAL leverages spatial attention to capture cross-pixel dependencies and perform alignment. This combination directly targets the two identified failure modes.

3.Evaluation covers both after-generation and in-generation settings across six geometric distortions, compares to multiple baselines (including MBRS, FIN, CIN, Gaussian Shading, GaussMarker), and includes ablations demonstrating the roles of CAS and spatial attention. Reported gains appear consistent across ACC and visual metrics.

**Weaknesses:**

1.Related work linkage. The related-works discussion, especially the in-generation watermarking subsection, feels somewhat abrupt and not tightly integrated with the proposed method. The manuscript should better explain how CASIAL relates to and differs from in-generation approaches.

2.Figure 3 readability. Figure 3 has small text and large whitespace; the layout could be optimized so readers can more quickly grasp the internal flow of the CAS block.

3.Insufficient detail for IAL / spatial attention. Although CAS is described in detail, the Methods section lacks a clear, concrete specification of IAL. The paper only notes in the Introduction that IAL uses spatial attention; Methods should explicitly describe the spatial-attention operations (architecture, formulas, how alignment is enforced).

4.Unclear integration with in-generation pipelines. The paper mainly describes application to after-generation watermarking. It does not clearly state how CASIAL is integrated into in-generation workflows (e.g., where CASIAL is inserted in an LDM pipeline, how it interacts with the VAE / denoiser / inversion process). This gap complicates the interpretation of the in-generation results.

**Questions:**

1.In Figure 2 the text states that “gradients through CAS are used only to update this block and are not propagated into the image feature extractor.” Why is the CAS gradient isolated in this way? Please explain the design motivation and implementation (e.g., stop-gradient, separate optimizers, or other mechanism) and discuss effects on training stability and coupling between CAS and the encoder.

2.What is the exact architecture of the spatial attention block? How are attention weights computed and applied? Are the spatial attention blocks inside CAS identical to those used in the encoder/decoder backbones, or are they different? Please provide a detailed description of the block, including formulas, layer types, and key hyperparameters.

3.The paper describes after-generation experiments in detail, but the in-generation application remains unclear. Where is CASIAL inserted in the LDM pipeline? How does CASIAL interact with the VAE latent, denoiser, and diffusion inversion during extraction? Please provide a concrete integration and implementation description.

4.Figure 5 suggests the method tends to embed watermark information into relatively smooth background regions. Is this tendency general across images and datasets? If smooth/background regions are cropped out, can the method still reliably decode from remaining regions? Please provide a statistical analysis (e.g., per-region contribution or decoding success conditioned on semantic region removal).

5.In Table 4, for the Jigsaw distortion, why does w/o SP (removing spatial attention blocks) still achieve near-100% accuracy? Can you explain this phenomenon or provide further ablations (e.g., varying jigsaw grid sizes, number of permutations, or seed sensitivity) to clarify when and why spatial attention is necessary?

---

### Official Review · Reviewer_7DgQ · 2025-11-02

**Soundness:** 2
**Presentation:** 2
**Contribution:** 2
**Rating:** 2
**Confidence:** 5

**Summary:**

In response to the weakness of current image watermarking models in against geometric distortions, this paper propose CASIAL, a geometric distortion–robust watermarking framework with cover image-aware message spreading (CAS) and invariance alignment learning (IAL). Although this method can withstand both region removal and desynchronization while preserving invisibility and fidelity, its contributions are limited. Specifically: The CAS technology is constrained by information length, and the design of the IAL mechanism remains under-explored.

**Strengths:**

See the Summary

**Weaknesses:**

(1)	Lack of comparison in model complexity, including: parameters, flops and etc.

(2)	Lack of introduction to the Spatial Attention Block

(3)	The design of the IAL mechanism is not well discussed

(4)	Why choose to use the GroupNorm at the end of the decoding process?

(5)	The watermark capacity tested in the paper is insufficient, limiting its practicality.

(6)	A careful analysis of the CAS design reveals that it will be constrained by information length

**Questions:**

See the weakness

---

### Meta-Review · Area_Chair_inrs · 2025-12-25

**Summary:**

The initial ratings of this paper are 2, 6, 4, and 4. The reviewers raised a lot of questions, and concerns, such as complexity, watermark capabilities, and details of IAL However, the authors did not submit rebuttal to address the reviewers’ concerns. Thus, AC recommends rejecting it.

**Reviewer Concerns:**

The authors did not submit rebuttal to address the reviewers’ concerns.

**Reviewer Scores:**

The authors did not submit rebuttal to address the reviewers’ concerns. No reviewers will change scores.

---

### Decision · Program_Chairs · 2026-01-26

Reject